# Activity driven sleep dynamics: A conceptual modeling study

Dmitry Postnov[1]*, Ksenia Merkulova[2], Oxana Semyachkina-Glushkovskaya[1], Jürgen Kurths[3,4]*

**1** Department of Biology, Saratov State University, Saratov, Russia, **2** Institute of Physics, Saratov State University, Saratov, Russia, **3** Res Inst Intelligent Complex Syst, Fudan University, Shanghai, China, **4** Potsdam Institute for Climate Impact Research, Telegrafenberg Potsdam, Germany

* postnovdmitry@googlemail.com (DP); kurths@pik-potsdam.de (JK)

## Abstract

The sleep state is traditionally defined as global, involving the entire body. Accordingly, a two-process mathematical model of the switch between sleep and wakefulness was proposed by Borbely more than 30 years ago and successfully used in deep modeling studies. However, in recent decades, new data have been obtained that call for a more complex understanding of sleep, both in spatial and temporal aspects. In particular, this concerns such phenomena as local sleep and short sleep episodes (naps). Here, we propose a mathematical model that gives the space for both local sleep phenomena and short sleep episodes. In doing so, we rely on the concept of a "sleep unit" and propose a phenomenological rate model of its activity. We also use a concept of a psycho-sensory drive to simulate various scenarios induced by external or internal stimuli. An essential element of our model is the representation of the background neural activity as a random signal, which reflects the real situation of overlapping multiple internal and external stimuli. The corresponding simulation runs confirm that the model reproduces both phenomena: local sleep and short nap episodes. As a particular but illustrative result, we demonstrate an example of involuntary global falling asleep under conditions of an overload of a separate sleep unit. In conclusion, we discuss the potential of our model and ways of its further quantitative parameterization.

## Introduction

Today, there is general agreement that the features of brain fluid movement are critically important for the process of clearing the brain of harmful metabolites and toxins. Several hypotheses have been proposed to interpret the data on fluid movement in the brain parenchyma [1–3], which recognize the important role of sleep in these processes. It has been shown that during deep sleep, the volume of extracellular space increases significantly. This increase occurs at a time of low neural activity characteristic of deep sleep (NREM), when slow wave activity (SWA) is recorded on

**Data availability statement:** All relevant data are contained in the manuscript and its supplementary materials. Specifically, our manuscript (i) does not contain experimental data, and (ii) all information necessary to reproduce our results (i.e., model equations and control parameter values) is included in the manuscript and supplementary materials.

**Funding:** DP and KM were supported by Russian Science Foundation, grant 25-15-00174. OSG and JK were supported by Russian Science Foundation, grant 23-75-30001. The funding organizations had no role in the study design, data collection and analysis, the decision to publish, or the preparation of the manuscript.

**Competing interests:** The authors have declared that no competing interests exist.

the electroencephalogram. Increased extracellular space contributes to the activation of the processes of clearing the brain of beta-amyloid, a risk factor for the development of Alzheimer's disease, and lack of sleep leads to its accumulation in the brain tissue of healthy people [4–7].

It is well known that when solving various problems, the brain demonstrates spatially nonuniform patterns of neuronal activity. Consequently, changes in blood flow and the associated movement of cerebrospinal fluid are also spatially nonuniform. This is the basis, for example, of transcranial stimulation techniques [8].

However, the state of sleep is usually defined as global, involving the entire body. In terms of the electrical activity of the brain, sleep is recorded as characteristic EEG patterns (slow wave activity, SWA), and their whole-brain synchrony is considered essential. Within the framework of this approach, it is natural to assume that the decision "to sleep or not to sleep" is made by the brain structures responsible for this - the sleep and wakefulness centers, while the rest of the brain simply obeys.

Thus, the concept of sleep as a spatially homogeneous state of the brain underlies a two-process mathematical model of switching between sleep and wakefulness [9], proposed by Borbely in 1982 and successfully used in numerous modeling studies.

However, during the past decades, a considerable amount of data has been accumulated that calls for a more complex understanding of sleep, both in spatial and temporal aspects. In particular, this concerns what is now known as "local sleep" and "power naps".

The study of the phenomenon of local sleep began with the discovery of unihemispheric sleep in marine mammals, birds, and reptiles [10–12]. The brain of reindeer has recently been reported to generate SWA during chewing: these animals can be awake and sleep at the same time [13].

In humans, it has been established that the left hemisphere of people functions differently from the right hemisphere when they sleep under unfamiliar conditions. It has been found that the amplitude of slow wave EEG activity can vary significantly for different locations in the cerebral cortex. Moreover, the increased activity of certain brain regions locally enhances SWA during sleep. It has been suggested that "sleep is a fundamental property of small viable neuronal/glial networks" [14].

It has also been shown that SWA activity rises locally and asynchronously across brain regions during non-rapid eye movement (NREM) sleep, with its regional distribution and amplitude being in part mediated by experience-dependent changes in synaptic strength. In addition, individual areas of low-frequency oscillations (<10 Hz) can also occur in REM sleep and during wakefulness. Local changes in SWA in posterior brain regions account for changes in consciousness during sleep [15]. Thus, it has been shown that "sleep homeostasis indeed has a local component" [16]. At the level of small cellular structures, a sleep-like behavior of rat cortical columns in vivo has been found [17,18]. In vitro studies provide further evidence for local sleep by demonstrating that cortical cultures exhibit short bursts of spontaneous activity closely resembling the sleep signature of an intact brain [19,20].

All of the above suggests that there must be signaling pathways for sleep control directed from the cellular structures of the cortex to the sleep centers. Indeed,

there is ample evidence that astrocytes are involved in controlling the need for sleep through several signaling pathways, adenosine-mediated [21–25] and adenosine-independent [26–28].

Based on all that, it has recently been suggested that the minimal functional unit of sleep - the "sleep unit", should be sought at the level of a neuro-glia-vascular unit. In the paper [29] the author expresses the opinion that "sleep is a property and is initiated by small groups of closely interconnected neurons". In the review [30] the role of astrocytes in sleep regulation is emphasized. On this basis the neuro-glia-vascular unit or rather their small ensembles are proposed as the sleep unit.

All of the above suggests that modern mathematical models should take into account spatial heterogeneity, such as local initiation of sleep, when describing the switch between sleep and wakefulness.

From a temporal perspective, the main challenge for a two-process model is that it does not assume "extra" short episodes of sleep. At the same time, such episodes often occur in blind people, when the action of the main zeitgeber is weakened [31–33]. In recent decades, the "power naps" resting technique has gained popularity [34–38]. Short naps have been shown to improve cognitive performance and have many benefits.

Finally, it has recently been shown that penguins prefer power naps [39]. It was found that penguins fell asleep more than 10,000 times a day for only 4 seconds, but they still managed to sleep for about 11 hours.

Returning to the two-process model, we note that all aspects described above do not have a clear link to the circadian rhythm and therefore require the inclusion of some other approaches of influencing sleep propensity in the model.

In our work, we propose a model hypothesis that combines the action of the systemic (central) sleep-wake switch and locations distributed throughout the brain, allowing for both local sleep and episodes of short sleep. In doing so, we focus more on the mutual action of the parts of the model than on their precise physiological interpretation.

First, we develop a phenomenological rate model of the activity of an individual sleep unit, which corresponds to either a separate neuro-glia-vascular unit or, more realistically, a small ensemble of them. Next, we introduce a psycho-sensory drive signal into the model following the ideas expressed in the work [40].

This allows us to simulate various scenarios dependent on external or internal stimuli. Finally, an essential property of our model is the presence of a background of neural activity in the form of a random signal, which reflects the real situation of a superposition of multiple internal and external stimuli controlling neural activity.

As a particular but indicative result, we demonstrate the performance of the local sleep model using the example of involuntary global falling asleep under conditions of overload of a separate sleep unit, if, in addition, the rest of the periphery receives a reduced level of stimulation. Then, using various combinations of control signals, we reproduce several scenarios of a listener's drowsiness during a boring afternoon lecture. In conclusion, we discuss the potential of our model and ways of its further quantitative parameterization.

## The model

### General architecture of the model

Our model should cover both systemic and local, but distributed throughout the brain, sleep control mechanisms. The parts included in the model and the ways of their interaction are shown schematically in Fig 1.

To describe the central sleep-wake switch, we use a well-studied and used in a number of applied problems two-process model in the version proposed in [41,42]. Among other ingredients, this model has two control parameters that characterize the contribution of lateral structures to the activity of sleep and wakefulness centers, respectively. We use these parameters as an interface for sleep drive and wake drive signals, which we believe are formed in the periphery and affect the moments of falling asleep and waking up.

To describe the formation of these signals, they reflect the state of a large array of sleep units, each of which corresponds to a small ensemble of closely located neuro-glia-vascular units interconnected and receiving the same (or very

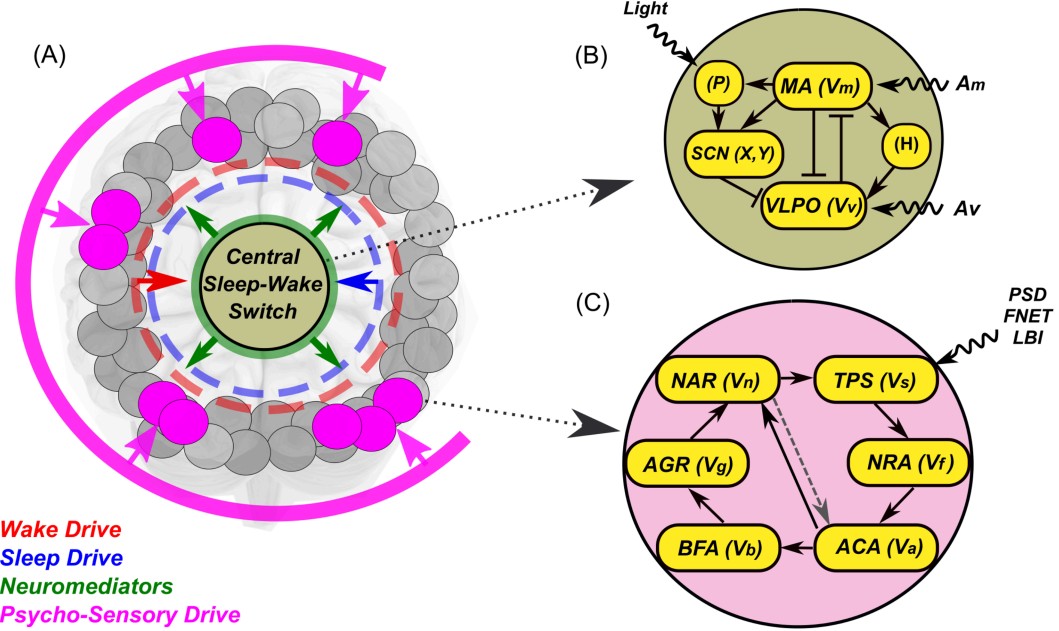

**Fig 1. Model flowcharts.** (A) The general structure of the model includes a large number of sleep units (filled circles) interacting with the central sleep-wake switch. Some of them (filled with magenta) form an active ensemble, receiving a common psycho-sensory drive. (B) The structure of the central sleep-wake switch model follows [41,42].The bar-headed and arrow-headed lines indicate inhibitory and excitatory connections, respectively. The abbreviations are: P - photoreceptors, SCN - suprachiasmatic nucleus of the hypothalamus, H - homeostatic process, MA - monoaminergic nuclei of the hypothalamus and brainstem, VLPO - ventrolateral preoptic nucleus of the hypothalamus. (C) The structure of the rate model of the individual sleep unit. The abbreviations are: TPS - total postsynaptic signal, NRA - neuronal rate activity, ACA - astrocyte calcium activity, BFA - blood flow activity, AGR - astrocytic glycogen reserves, NAR - noradrenaline release, PSD - psycho-sensory drive, FNET - inputs from functional network, LBI - lateral background inputs. For details see text.

similar) external stimuli. In turn, the central switch affects the state of the periphery through a number of neurotransmitters, of which noradrenaline can be regarded as the main one. An important element of the model is the presence of a psycho-sensory drive signal in the spirit of [40]. This empirically determined signal carries information about the scenario of events, which is determined by both external conditions and intracerebral settings.

Below we describe in more detail the specific implementation of each of the above parts of the model.

## A model of the central sleep-wake switch

Fig 1B shows the structure of the sleep-wake switch model [41,42]. Since we did not change this part of our model compared to the original publication, we placed its detailed description in S1 Appendix. Here we explain its operation as much as is necessary for understanding our work and results.

This neural mass model describes processes at the level of neuronal populations. The basis of the model is two mutually inhibitory centers: wakefulness and sleep, which are physiologically represented by the monoaminergic nuclei of the brainstem (MA) and the ventrolateral preoptic nucleus of the hypothalamus (VLPO), respectively. The activity of these nuclei is described by the average voltage of the neuronal populations and is quantitatively represented by $V_m$ and $V_v$, respectively. The state of wakefulness is registered when $V_m$ exceeds a given threshold value, otherwise the system is considered to be in a sleep state. The VLPO and MA nuclei form a bistable system, which, in the absence of external influence, remains in one of the two stable states for an indefinite period.

Switching between states occurs using circadian (SCN, variables $X$ and $Y$) and homeostatic (variable $H$) processes.

The homeostatic process $H$ is completely controlled by the activity of the wakefulness center and reflects the degree of sleep inclination, which increases during wakefulness and decreases during sleep.

The circadian process is carried away by the daily 24-hour rhythm and reacts to changes in environmental illumination. This is done with the help of special light-sensitive ganglion cells in the retina of the eyes - photoreceptors (variable $P$).

The equations for the variables $V_m$ and $V_v$ include the control parameters, $A_m$ and $A_v$, respectively, which have the meaning of the total signals coming to the wakefulness and sleep centers from all other parts of the brain. This gives us a natural interface for connecting with other parts of our model.

### Psycho-sensory drive

The term "psycho-sensory drive" (PSD) was proposed in [40]. PSD is a signal that generally describes the effect of a variety of factors, both internal (psycho-emotional state, imaginary situations, etc.) and external ones (perception of audiovisual information, food intake, etc.). While it is impossible to deny the significant impact of PSD on the sleep-wake regime, it is difficult, if not impossible, to formally describe its mechanisms, and even more so - to reasonably set a quantitative measure. Indeed, how can one measure, for example, the impact of anxious thoughts or emotional tension before a socially significant event that prevents a person from falling asleep? Nevertheless, taking PSD into account in modeling studies seems promising, since it allows one to simulate typical socially significant factors and significantly bring the calculation results closer to real situations. In our work, we describe an attempt to take PSD into account at the basic level.

From a formal point of view, the PSD signal can be introduced into an N-dimensional mathematical model as, for example, an N-component signal $PSD(t, i)$, $i = 1 \dots N$, which is added and/or multiplicatively included into the right-hand side of each equation in the model. When added, the PSD signal increases or decreases the rate of change of the variable. When included multiplicatively, the PSD signal acts as a modulator, enhancing or weakening the effect of other factors. Thus, there can be quite a few formally acceptable ways of including the PSD in the equations.

In solving this problem, we first tested the introduction of a PSD into the central switch model [41,42]. It was necessary to artificially induce daytime sleep, which is not realized within the framework of the normal operating mode of this model. As already mentioned, the equations for variables of the sleep center $V_v$ and wake center $V_m$ contain the parameters $A_v$ and $A_m$, which describe the action of other (not included in the sleep-wake switch) neuronal nuclei. By making these parameters time-dependent in the same way as the PSD signal changes, we obtain its two components.

In the equation for the homeostatic variable $H$, the important parameter is $\tau_H$, which generally represents the combined effect of basic factors, including the intensity of activity during the day, the quality of sleep at night, etc. It seems reasonable to make this value partially dependent on the PSD signal and, thus, influence the rate of fatigue and rest.

At the same time, the circadian rhythm, although subject to external influences, is much more stable and is normally synchronized with the 24-hour light rhythm. For this reason, we assume that PSD does not appear in the equations for the circadian oscillator variables $X, Y$ and the photoreceptor activity $P$.

Manipulating the PSD signal introduced in this way allows us to control the dynamics of the sleep-wake switch. Fig 2 presents the effect of a sequential 15-minute decrease in the parameter $A_m$ (at 15:00 by 25%), and 2 hours later - an increase of the same duration in $A_v$ by 15%. As can be seen, in response to these effects, the system obediently "falls asleep" and "wakes up", simulating a daytime sleep lasting 2 hours.

For the sleep unit model, it was decided to introduce the PSD signal in a similar way, adding it to the equation describing the set of different signals coming to the neuron. This is described in more detail in the next subsection.

### Sleep unit

A mathematical model of the minimal functional unit of sleep is developed to implement the working hypothesis of the existence of a sleep-like state, a state of reduced activity with reduced sensitivity to stimuli in one individual neuro-glia-vascular unit (NGVU). Namely, such a model should describe: (i) changes in the level of neuron activity, when the

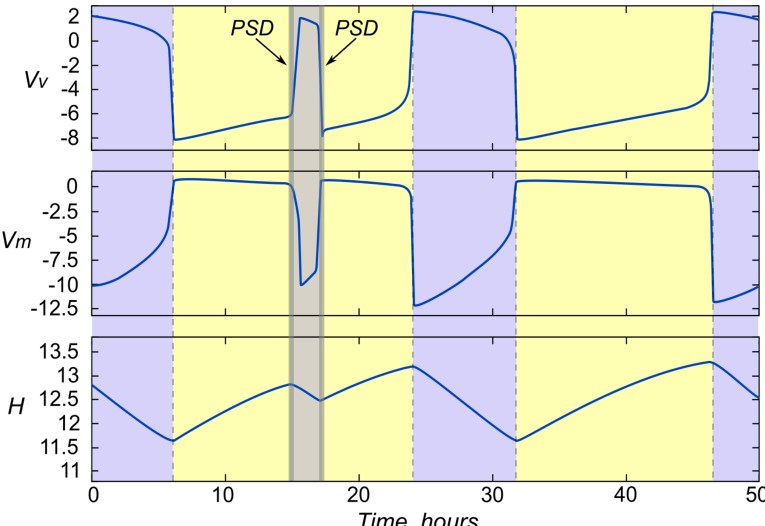

**Fig 2. PSD in global model.** Time courses of variables $V_v$, $V_m$ and $H$ with a 15-minute change of the parameters $A_m$ (at 15:00 by 25%) and $A_v$ (at 17:00 by 15%). Sleep state is shown in light purple. The model demonstrates that stimulation causes daytime sleep lasting 2 hours.

level of stimulation from systemic mechanisms changes; (ii) the proportional response of the neuron to stimuli in the physiologically normal range; (iii) temporary "switching off" of the NGVU during prolonged hyperactivity of the neuron, when the current expenditure of metabolic energy (in the form of oxygen, glucose, astrocytic glycogen) exceeds their supply with the blood flow.

A diagram reflecting the structure of the model is given in Fig 3.

The activity of a neuron is described by the firing rate $V_f$ using an ordinary differential equation:

$$\tau_f \frac{dV_f}{dt} = F_{inf} - V_f,$$ (1)

where $F_{inf}$ specifies the neuron activity depending on the values of other model variables. This function has a range from 0 to 2 and describes the following expression:

$$F_{inf} = k_1 k_2 (0.5 + (1 - 0.5)k_3) V_s V_n.$$ (2)

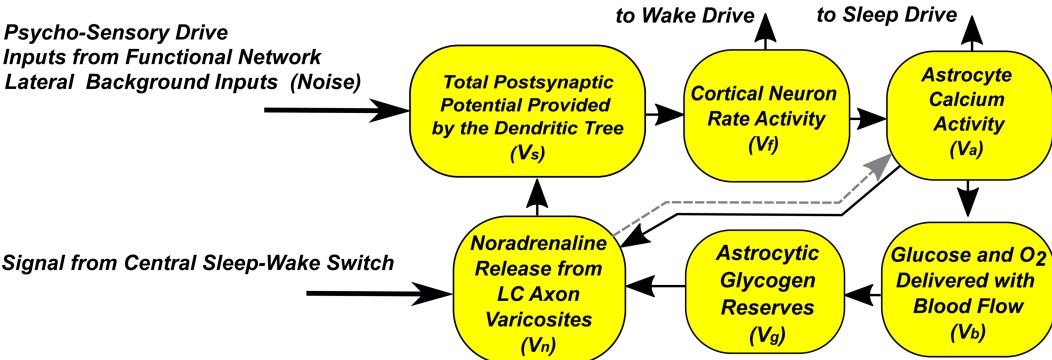

**Fig 3. Block diagram of a single Sleep Unit model.**

Here, the frequency of neural activity changes in response to a change in $V_s$, which describes the total postsynaptic potential provided by the dendritic tree. The degree of this influence depends on the rate of noradrenaline release $V_n$, as well as the factors $k_1, k_2$, and $k_3$, described below.

$V_s$, in turn, is formed by three signals: the psycho-sensory drive $PSD(t)$, the excitatory signals of other NGVUs included in the same functional ensemble $V_{fnet}$, and the background noise-like signal $V_r^2$, which describes the contribution of the rest of the brain and the NGVU's own spontaneous activity. Thus, the equation for $V_s$ takes the following form:

$$\tau_s \frac{dV_s}{dt} = \{PSD(t) + V_{fnet} + V_r^2\} - V_s, \tag{3}$$

where $PSD(t)$ depends on time as chosen for a particular simulation run,

$$V_{fnet} = k_{net}\frac{1}{N}\sum V_{f_i}^i, \quad i = 1 \dots N, \tag{4}$$

$$\tau_r \frac{dV_r}{dt} = (1 - V_r) + D\xi(t). \tag{5}$$

In these expressions $\xi(t)$ is white Gaussian noise; $D$ is the noise intensity, and $k_{net}$ rates the excitatory connections within the functional ensemble of sleep units.

From the system-level mechanisms side, the sensitivity and activity of the neuron are modulated by secretion of noradrenaline. In the model, this process is represented by $V_n$, through which local NGVU is linked to the global dynamics of "sleep-wakefulness" [43,44]. According to modern concepts, the process of noradrenaline release from the axon varicocites of the adrenergic neuron depends on the state of NGVU itself. In particular, it is supported by gliotransmitters released by the astrocyte, the synthesis of which, in turn, depends on the supply of astrocytic glycogen [45–47].

The noradrenergic neurons are located in the locus coeruleus (LC) neuronal nucleus, which is part of the wakefulness center. The main functions of LC are maintaining wakefulness and facilitating awakening from sensory stimuli [48]. On average, during quiet wakefulness, the locus coeruleus generates electrical impulses with a frequency of 1-3 Hz, and during sleep - with a frequency of 0.5-0.6 Hz [49,50]. Within the framework of the model, we assume that in a state of sleep, the release of noradrenaline is 20% of the maximum during wakefulness. The dynamics of $V_n$ is controlled by the global model as follows: according to [41,42], the state of wakefulness is diagnosed, when the variable of the wakefulness center exceeds a given threshold value ($V_m > V_{th}$), otherwise the state of the model corresponds to sleep. If we introduce an auxiliary quantity $S$, which takes the value of one when ($V_m > V_{th}$) and zero otherwise, then we can define $V_n$ as follows:

$$V_n = k_0 k_3 (0.2 + (1 - 0.2)S). \tag{6}$$

The action of regulatory pathways in NGVU is taken into account in the form of coefficients that reduce the release of noradrenaline or weaken the efficiency of synaptic transmission in relation to the normally active mode:

$$k_0 = min\{ (k_0^{min} + k_0^\Delta V_a), 1.0\}, \tag{7}$$

$$k_1 = min\{ (k_1^{min} + k_1^\Delta V_n), 1.0\}, \tag{8}$$

$$k_2 = min\{ (k_2^{min} + k_2^\Delta V_a), 1.0\}, \tag{9}$$

$$k_3 = 1 \quad if(V_g > 0.5) \quad and \quad k_3 = 0 \quad otherwise. \tag{10}$$

Here $k_0$ describes the attenuation of noradrenaline release with decreased astrocyte activity and, consequently, gliotransmission; $k_1$ characterizes the attenuation of synaptic transmission with low noradrenaline levels; $k_2$ takes into

account the attenuation of synaptic transmission with an inactive astrocyte due to an increase in the extracellular space and changes in the release of gliotransmitters. Finally, $k_3$ takes the values of 1 or 0 depending on the sufficiency of astrocytic glycogen stores to support processes in both the astrocyte and the neuron, see the description of the equation for $V_g$ below.

In the Introduction we already mentioned the importance of astrocytes in synaptic signal transmission and energy supply of NGVU. In our model, we take into account two important facts: (i) the supply of nutrients necessary for neuron activity occurs through the blood flow, the activity of which is regulated by the astrocyte; (ii) regulation of the blood vessel diameter in accordance with the current needs of the neuron is one of the most important functions of the astrocyte in NGVU. Vasoactive substances released from the astrocyte terminals cause vasodilation or constriction of blood vessels. When a neuron's activity increases, blood flow to the area increases, providing the neuron with sufficient nutrients. This increase in local blood flow in response to neuronal activity is called functional hyperemia [51,52] and was described in a number of modeling studies [53–55].

Here we use a simplified description of the dependence of the blood flow intensity $V_b$ on the activity of the neuron $V_f$ and the astrocyte $V_a$ in the following form:

$$\tau_b \frac{dV_b}{dt} = 5B_F(1 - V_b) - (V_f + V_a), \tag{11}$$

where

$$B_F = \frac{2 + exp(-4(V_f - 1))}{exp(4(V_f - 1)) + exp(-4(V_f - 1))}. \tag{12}$$

The function $B_F$ takes into account the fact that the intensity of blood flow increases as the frequency of spike generation increases. However, when the frequency increases significantly, blood flow paradoxically decreases [56]. This behavior is probably related to the high level of astrocytic endfoot calcium, which was demonstrated in [57].

A number of important processes in NGVU, such as gliotransmission and maintenance of neuron function, rely on astrocytic glycogen reserve, into which part of the incoming glucose is converted. In total, the amount of glycogen stored in an astrocyte is small and amounts to no more than 6% of the amount of glucose. The main consumption of glycogen occurs during energy-dependent processes in the neuron [45,58].

Note that to maintain these processes, it is not the total amount of glycogen stored by the astrocyte that is important, but its availability $V_g$. For this reason, we model the dynamics with a bistable oscillator, considering that this is a simplified one-dimensional description of a complex set of processes of glycogen formation, its breakdown and diffusion to the sites of consumption. Bistability here is an image of processes with memory, in the sense that when glycogen reserves are exhausted, it becomes available again only after a certain time, which is necessary for its resynthesis. Mathematically, this is described as follows:

$$\tau_g \frac{dV_g}{dt} = \Psi_G + \Psi_B, \tag{13}$$

where

$$\Psi_G = 0.6\sqrt{3}(2V_g - 1 - (2V_g - 1)^3), \tag{14}$$

$$\Psi_B = 2V_b - 0.5. \tag{15}$$

 

If blood flow decreases, energy for neuron activity begins to flow in the form of astrocytic glycogen. That is, the value of $V_g$ has two stable levels: high, if metabolic reserves are sufficient, and low, if they are significantly reduced. This bistability is specified by the functions $\Psi_G$ and $\Psi_B$. Also, we take into account that during sleep the glycogen level increases (is restored).

The degree of astrocyte activity is traditionally estimated as the intensity of calcium waves in it. This activity generally follows the activity of the neuron, but at the same time has a slower and smoother dynamics in time. This is due to the characteristic time of the astrocyte response to external influence, which is about few seconds, compared to fractions of a second, on which the firing rate of the neuron can be calculated. We describe the activity of the astrocyte in the most simplified way as follows:

$$\tau_a \frac{dV_a}{dt} = V_f - V_a. \tag{16}$$

Note that the variables and functions of the model are normalized in such a way that in the normal daily activity mode, most of them are equal to one.

## Numerical methods and values of control parameters

Numerical integration of model equations was carried out by the 4th order Runge-Kutta method, adapted for solving stochastic differential equations. The basic set of model parameters is as follows: $\tau_f$ = 0.1 / 3600; $\tau_a$ = 1 / 3600; $\tau_b$ = 2000 / 3600; $\tau_g$ = 1 / 3600; $\tau_s$ = 0.05 / 3600; $\tau_r$ = 1; $D$ = 50; $k_0^{min}$ = 0.8; $k_1^{min}$ = 0.8; $k_2^{min}$ = 0.8; $k_0^\Delta$ = 0.2; $k_1^\Delta$ = 0.2; $k_2^\Delta$ = 0.2.

The first six parameters set the characteristic times of the described processes. They are easy to estimate in relation to the parameters of the central sleep-wake switch model, where time is measured in hours. Taking into account the division by 3600, the given numbers correspond to the interval from 0.1s for the neuronal reaction to tens of minutes for the reduction of glycogen stores with insufficient blood flow.

The remaining 7 parameters were selected empirically so that the model behavior was reasonable from the point of view of the implemented pathways. In fact, we simply set the maximum relative change in the $k_0$, $k_1$, and $k_2$ as 20% of its initial value.

# Results

Since we propose the model structure from general considerations, although taking into account physiology, a numerical experiment is necessary to evaluate the effectiveness of the proposed structures. Also, it is helpful to clarify the requirements for those connections that are introduced in a highly simplified manner, due to the lack of knowledge about the characteristics and parameters of individual physiological signaling pathways.

First, we tested the behavior of a single sleep unit in the absence of connections with other sleep units in a functional ensemble and with a fixed value of the noradrenaline level. This is not very realistic from a physiological point of view, but it allows us to see how the mechanisms embedded in the model work.

## Operation of single sleep unit model

In this subsection, we show that a single sleep unit works as expected, i.e., briefly "switches off", when the stimulus intensity is too high to replenish energy resources.

In the Fig 4, we demonstrate the effect of increased psycho-sensory drive on a single sleep unit. During general wakefulness, at time $t \in (1, 2)$, there was a sharp increase in the external stimulation $V_s$, which in its intensity is approximately 60% higher than the average daily psycho-sensory drive. In response to the increase in external stimulation, the frequency of spike generation by the sleep unit neuron sharply increases (by about half), and, as a result, astrocytic activity $V_a$ increases.

 

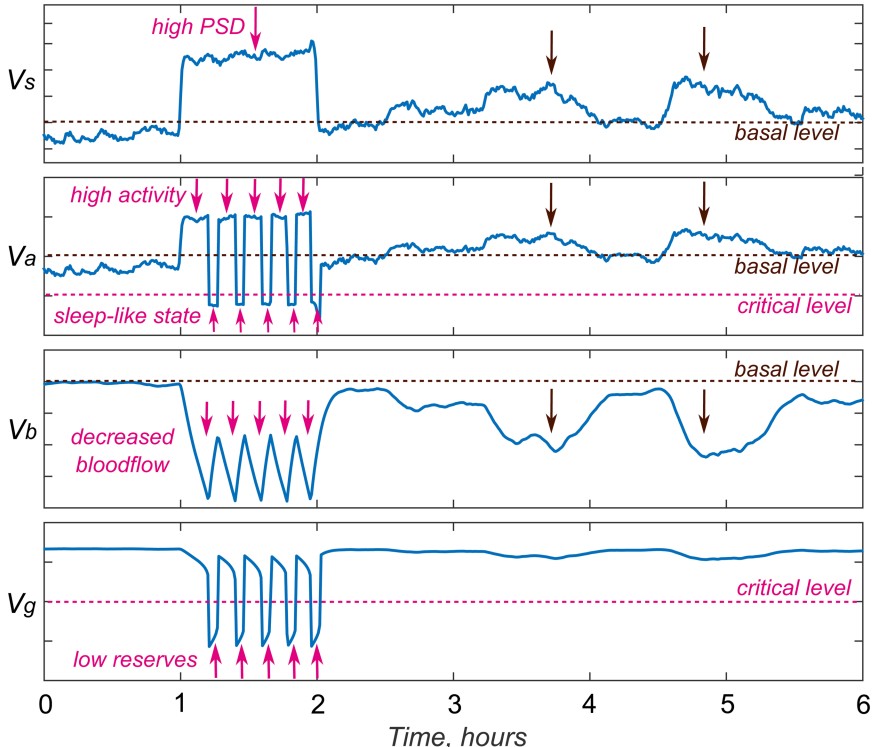

**Fig 4**. **On the behavior of an individual sleep unit.** Graphs of selected model variables show the response to an excessively strong stimulus (red arrows) and to a "normally high" noise-like stimulus (black arrows). For further details see text.

Increased neuron activity increases energy expenditure in the NGVU, which, in turn, leads to a decrease in glucose and oxygen coming from the blood flow to "protect" the neuron from "overload". As mentioned above, energy comes to the neuron not only through the blood flow $V_b$ , but also in the form of astrocytic glycogen, which begins to be actively spent on maintaining the neuron that has lost its energy resources that is, the supply of glycogen $V_g$ is depleted. After the depletion of astrocytic glycogen reserves, the neuron's activity and its sensitivity to external stimuli drop sharply. In this case, we can talk about a sleep-like state. During this time, the blood flow increases, the glycogen energy reserves are replenished, and the neuron "wakes up" because there is energy for its activity and the external stimulus is still strong.

In the case where PSD increases not so much (by 25-30%), there is also an increase in neuronal and astrocytic activity, a decrease in the supply of energy reserves from the blood flow, but glycogen is practically not consumed. It is believed that NGVU operates in a mode of increased, but not excessive activity. These situations are demonstrated in the Fig 4 by black arrows.

This result is consistent with the intracellular recordings, which show that a significant stimulus is needed to activate an individual neuron [59]. However, if we consider a population of neurons, a response to a weak signal will also be observed. This is due to the fact that a large number of neurons in the population receive the signal at once. This will be discussed in more detail in the next subsection.

## Cooperative behavior of coupled sleep units

In this subsection, we analyze the situation when a sleep unit is part of a functional network - a set of similar units that we believe are interconnected by mutually excitatory connections, with some elements of such a network receiving the same psycho-sensory drive.

We consider the essential control parameters to be (i) the degree of connection $k_{net}$ of individual sleep units, and (ii) what percentage of the total number of network elements receives the *PSD* signal, $PSD_{inputs}$. Note that, as in the previous subsection, each sleep unit also receives a noise-like input signal reflecting both the influence of the environment and its own spontaneous activity.

Simulation runs were performed on the time interval from 0 to 6 hours, with the PSD signal applied to the network in the time interval from 2 to 4 hours. Different sizes of the functional network were tested. For the illustration below, we chose a network consisting of $N = 100$ sleep units. The results are presented in Fig 5. Panel A of the figure shows a sketch of the regime map on the two-parameter plane. We identified three regimes:

1) At small values of $k_{net}$ and/or $PSD_{inputs}$, the PSD signal causes an increase in activity, which we estimated by calculating the ensemble mean $a_{mean} = (\sum V_a^i / N)$, but does not lead to the dips caused by the sleep-like state, as discussed in the previous subsection.

2) At large values of $k_{net}$ and/or $PSD_{inputs}$, such dips in activity were well defined and regular.

3) In the region of intermediate values of $k_{net}$ and/or $PSD_{inputs}$, the dips in the activity graph were irregular and did not repeat from one simulation run to another.

Panels B and C illustrate how the graph of $a_{mean}$ changes, when moving along the dotted lines on the mode map.

Of course, both the smooth activity graph for small $k_{net}$ and/or $PSD_{inputs}$ and the sequence of almost periodic pulses do not look very similar to real activity of the cellular structures of the brain parenchyma. This is a consequence of the fact that we have unrealistically set the psycho-sensory drive time dependence. However, this computational experiment allowed us to estimate how a functional ensemble of sleep units of different sizes behaves and to find a way to take this into account without integrating models from an indefinitely large number of elements.

Our interpretation of the results obtained is as follows: when we create an ensemble even from several elements, it acquires new properties compared to a single sleep unit. Namely, the states of both general high activity and general

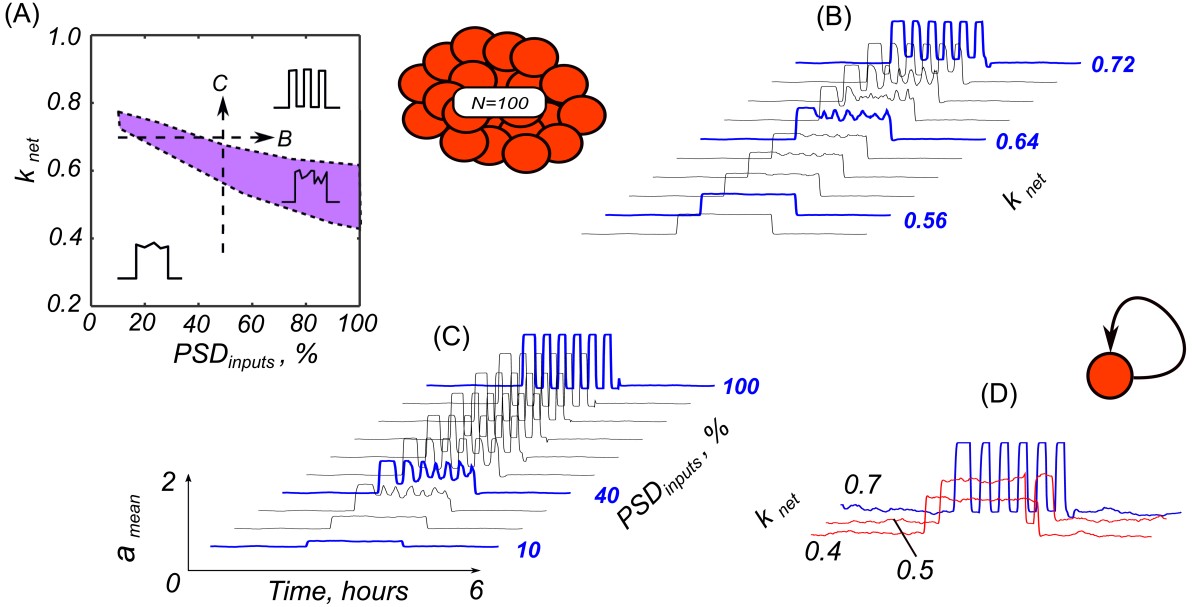

**Fig 5**. **Averaged activity of an ensemble.** An ensemble include of 100 sleep units as a function of the parameters $k_{net}$ and $PSD_{inputs}$, which specify the strength of excitatory connections within the ensemble and the percentage of units receiving an external PSD signal, respectively. (A) Sketch of the diagram showing the regions of the main modes. (B) and (C) The change of mode along the dashed lines B and C in panel A, respectively. (D) A similar signal obtained by replacing the ensemble with a single sleep unit equipped with positive feedback. For further details see the text.

low activity become more stable and tend to synchronize. Indeed, in different connected sleep units, the PSD signal acts against the background of their own noise-like signal, which is different in each element of the ensemble. However, if one of the elements has entered a sleep-like state of low activity, then its own activity has lowered the overall stimulus for all the others, and made the same transition more likely for the remaining units. Accordingly, turning off the next unit further lowers the overall level of the stimulus signal. The process develops like an avalanche, and soon the entire ensemble enters a state of low activity. Similar events occur during the reverse transition.

In essence, the mechanism described above is a very basic action of a positive feedback. Since this is so, we can try to model the dynamics of the ensemble using just one element, equipped with a positive feedback with a suitable delay $\tau_{fnet}$:

$$\tau_{fnet}\frac{dV_{fnet}}{dt} = V_f - V_{fnet}. \tag{17}$$

The result of such a replacement is shown in Fig 5D. We find that the behavior of a single sleep unit with the added positive feedback imitates the behavior of the ensemble very well. This allows us to significantly reduce the model, and most importantly - eliminates the need to determine how many sleep units we should include in the network.

## Local + Global model

The results described above allow us to consider episodes of global sleep under the assumption that the sleep centers are affected by signals from only two clusters of neurons, one of which represents an active functional network engaged in solving the current task, and the second - the rest of the brain, creating a noise-like background during the processing of various external and internal stimuli. At the same time, both neural clusters contribute both to the maintenance of the awake state and to the accumulation of sleep drive. The block diagram of the model constructed on the basis of these assumptions is shown in Fig 6. Assuming that wake drive is generated by neurons and sleep drive is generated by astrocytes [21,60–62], we introduce two new variables, $V_{wd}$ and $V_{sd}$, respectively:

$$\tau_{wd}\frac{dV_{wd}}{dt} = V_f - V_{wd}, \tag{18}$$

$$\tau_{sd}\frac{dV_{sd}}{dt} = 2V_a - V_{sd}, \tag{19}$$

where $\tau_{wd}$ and $\tau_{sd}$ define the characteristic rise/fall times of the variables. It would be very straightforward to add these variables directly to the equations for $V_m$ and $V_v$ using the parameters $A_m$ and $A_v$. According to the original model [41, 42], these parameters contain signals from all neural nuclei external to the sleep-wake switch. It is difficult to specify in what proportions any of the discussed signals reach certain neural nuclei. Therefore, we make $A_m$ and $A_v$ dependent on all these signals, which include: neural activity and adenosine signaling from astrocytes of the active functional network, noise-like background activity of all other neural structures, and psycho-sensory drive. These signals are formed as follows:

$$A_m = A_{m0}\{1 + k_{m1}V_{wd} - k_{m2}V_{sd} + k_{m3}\xi(t) + k_{m4}PSD(t)\}, \tag{20}$$

$$A_v = A_{v0}\{1 + k_{v1}V_{wd} - k_{v2}V_{sd} + k_{v3}\xi(t) + k_{v4}PSD(t)\}, \tag{21}$$

where $k_{m1}$ and $k_{v1}$ represent the contribution of neural activity, $k_{m2}$ and $k_{v2}$ represent the sleep drive generated by astrocytes, $k_{m3}$ and $k_{v3}$ specify the contribution of noise-like activity from the rest of the neural structures of the brain $\xi(t)$, and $k_{m4}$ and $k_{v4}$ scale the possible direct contribution of the PSD signal. Since four pairs of parameters give a large set of combinations, which is not easy for computational study. In the first guessing step, we set $k_{m1} = k_{v1} = k_{m2} = k_{v2} = k_{mv}$, and

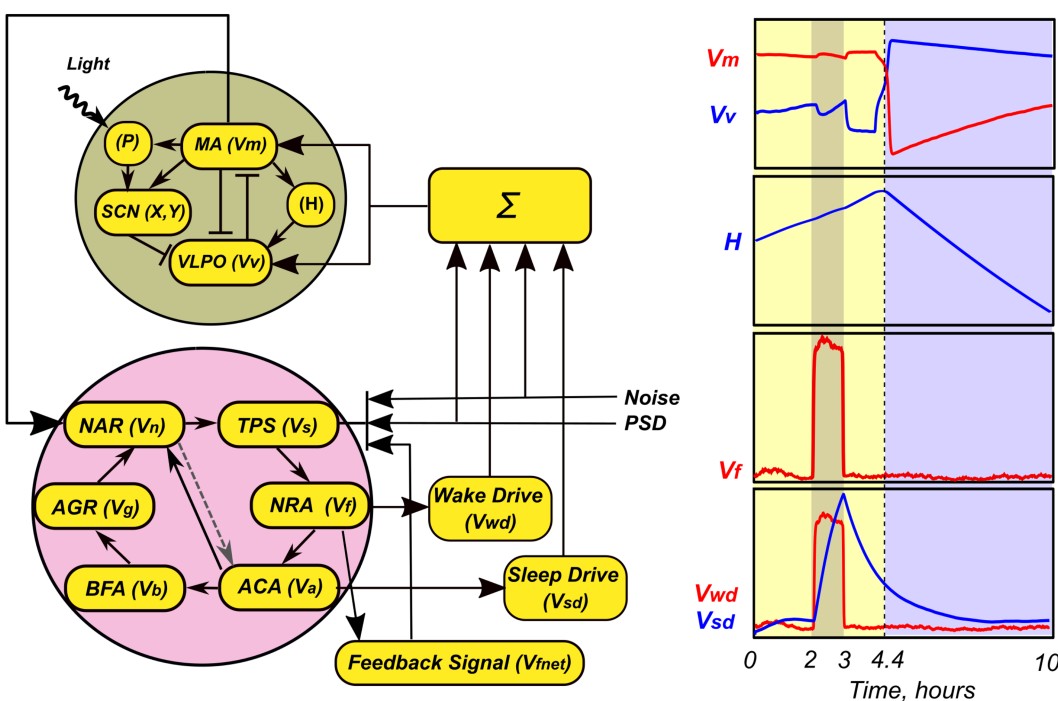

**Fig 6. Scenario modeling.** Block diagram of the relationship between the parts of the model, describing the global (top) and local (bottom) parts. On the right are the time dependencies of the variables of the original model and the local model. See text for details.

also $k_{m3} = k_{v3} = k_{mv3}$ and $k_{m4} = k_{v4} = k_{mv4}$. Such a symmetrical choice, as our simulations showed, provides that the work of the whole model is reasonable from a physiological point of view.

Also, the variable $V_n$ of the sleep unit model is now controlled by $V_m$ in the global sleep-wake switch model. Recall that the physiological prototype of this communication pathway is the production and distribution of noradrenaline by the LC neuronal nucleus, which is part of the structures responsible for maintaining the state of wakefulness.

The operation of such a unified model is illustrated in Fig 6 on the right. For this illustration, $k_{mv} = 0.15$, $k_{mv3} = 0$, $k_{mv4} = 0$. The component $PSD(t)$ has a moderately high value during one hour (from 2 to 3), and $k_{mv4} = 0.35$ with a shift of one hour. As a result, the generated wake drive ($V_{wd}$, red curve) and sleep drive ($V_{sd}$, blue curve) signals have an increased value, but different shapes. Recall that $V_{wd}$ reflects neural activity and quickly decreases with its cessation. In turn, $V_{sd}$ has physiological prototypes in the form of formation and accumulation of the neurotransmitter adenosine in brain tissues and is therefore characterized by a significantly larger time constant. In response to the action of $V_{wd}$ and $V_{sd}$, the graphs of the variables $V_m$ and $V_v$ change noticeably, while the variable $H$ reacts weakly. As a result, the transition to the sleep state occurs with a very small shift relative to the case when the PSD signal remained at a low level. Thus, in the shown case, the effect of the local part of the model on the global sleep-wake switch has a discernible effect, but does not disrupt its operation.

## Modeling of nap scenarios

In this subsection, by manipulating the two-component signal of psycho-sensory drive, we reproduce various variants of the familiar situation of a short sleep of a listener during a lecture or presentation. In Fig 7 we present the most illustrative cases in the form of curves of the variables $V_v$ and $H$. We show that the dynamics of the model reproduces the possible spectrum of responses, from successfully fighting sleep to deep sleep, when a significant external stimulus (noise at

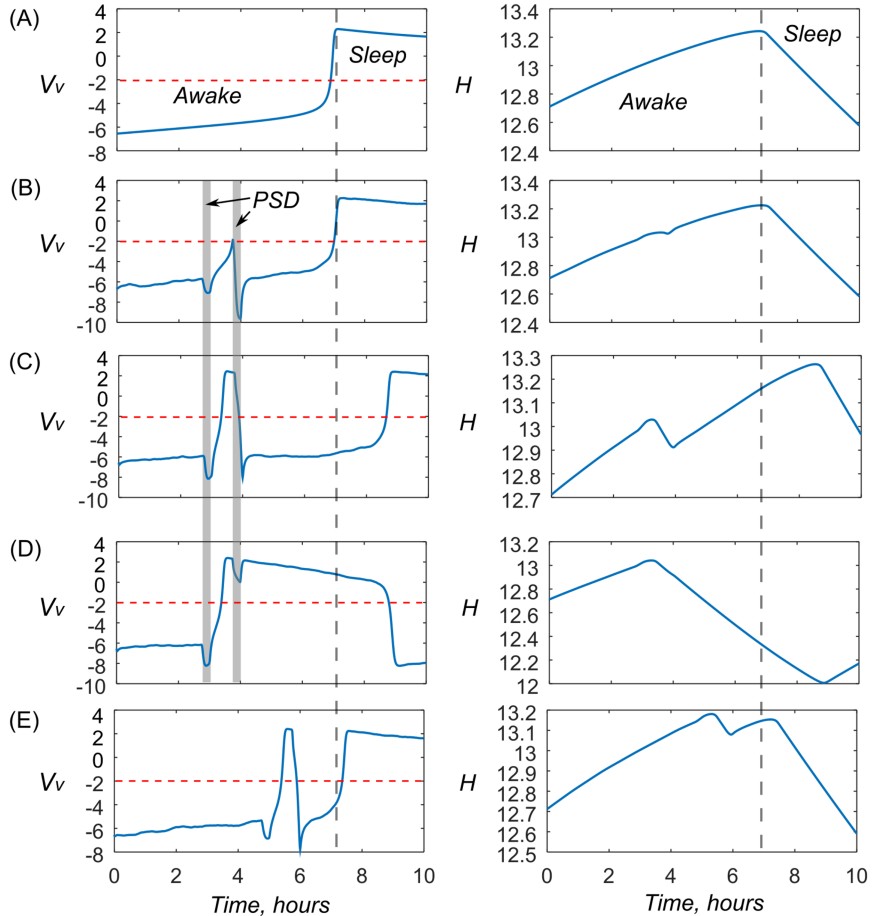

**Fig 7. Scenario modeling. Time realizations of variables $V_v$ and $H$.** Parameter values. (A) $k_{mv} = 0.0$, $k_{mv3} = 0$, $k_{mv4} = 0$; (B) $k_{mv} = 0.03$, $k_{mv3} = 0.02$, $k_{mv4} = 0.15$; (C) $k_{mv} = 0.06$, $k_{mv3} = 0.02$, $k_{mv4} = 0.15$; (D) $k_{mv} = 0.06$, $k_{mv3} = 0.02$, $k_{mv4} = 0.1$; (E) as for panel B, but PSD signal was applied one hour later, from $t = 4.75$ till $t = 5.0$.

the end of the lecture) is required to wake up. The figure contains 5 rows, labeled from A to E. The parameter values are given in the caption. Row A shows the unforced behavior of the global sleep-wake switch. The transition to sleep occurs at approximately $t \approx 7.5$, this moment is marked by the vertical dotted line. The horizontal red dotted line shows the level above which the variable $V_v$ means the transition to sleep.

For rows B, C and D the PSD signal consists of two components. The first, $PSD(t)$, included in the Eq (3), level 0.7 in the time interval $t = 2.75 \ldots 3.0$ and is zero in other cases, and the second, in the Eqs (20) and (21) is delayed by 1 hour and has the level $k_{mv3}$. According to the meaning of the simulated scenario, the signal $PSD(t)$ corresponded to a high load on perception, which simultaneously generated high neural activity and an increase in sleep drive. The PSD component in the Eqs (20) and (21) plays the role of a non-specific awakening stimulus, for example, noise at the end of a lecture. The action time of both components is shown as vertical gray stripes.

Row B illustrates the case when the action of psycho-sensory drive at $k_{mv} = 0.03$ changed the time dependence of $V_v$, but the transition to sleep does not occur. This event of short-term sleepiness has virtually no effect on the time of transition to long sleep (vertical dotted line), which is also evident from the graph of variable $H$, which remains virtually unchanged.

Row C is calculated for a stronger action of $V_{wd}$ and $V_{sd}$, at $k_{mv} = 0.06$. The exit of the blue curve to the upper half of the graph corresponds to the transition to sleep, but the subsequent second component of PSD stops this sleep. The situation corresponds to what is often observed at public lectures, when some people, sincerely trying to understand what the speaker is saying (overload of the functional network!) still fall asleep, but wake up from the change in the noise level at the end of the lecture. Note that a noticeable section with a negative slope has appeared on the graph of the $H$ variable and, as a consequence, the time of subsequent falling asleep has also shifted. This is completely consistent with the meaning of the [41,42] model, where an increase in the homeostatic variable $H$ describes the accumulation of fatigue, and a decrease - the restoration of strength.

Series D is calculated with the same parameter values as C, but the second component of the PSD is weakened, $k_{mv3} = 0.1$. As can be seen, there is no awakening at the end of the lecture (everyone quietly got up and left), and the accidentally occurring sleep continues, disrupting the normal phases of the sleep-wake cycle.

Finally, series E illustrates the fact that the result of the PSD action depends on the chosen moment. Delayed application of the same stimuli as for case B causes a transition to a short sleep as in C. However, the main episode of sleep shifts insignificantly.

To sum up, the scenarios described above seem quite realistic and do not contradict current ideas about the phenomenon of short sleep.

## Discussion

We have here for the first time proposed a way to take into account important phenomena and mechanisms in a model study, such as local sleep and short sleep episodes, by substantially extending the known model of two processes.

At the local (cellular) level, we have developed an extremely reduced, but physiologically explainable model of the reaction of a neuron-glia-vascular unit to both normal-high neuron activity and to "overload", which causes its temporary transition to a state of low sensitivity and low intrinsic activity. For ensembles of neuro-glia-vascular units, this transition can be regarded as an analogue of sleep at the cellular level [17,18].

There is a direct connection with the processes of clearing the brain parenchyma from harmful metabolites [4], since the state of low activity of the neuro-glia-vascular units corresponds to a significant increase in the extracellular volume, which significantly accelerates molecular transport [30]. This is one of the promising future applications of this proposed model.

For the model of the global sleep-wake switch, we used the [41,42] model without changes, but generalized the use of the control parameters available in it as an "interface" for interaction with the periphery. Detailing the pathways of this interaction is a separate difficult task, for which there is currently insufficient data. We relied on very basic facts, for example, that the sleep drive signaling pathway is at least partially mediated by adenosine-dependent astrocyte signaling pathways, as well as on the idea of psycho-sensory drive as a complex, both internal and external, source of activity maintenance [40].

We believe that the strengths of our work are as follows.

1. The two-process model taken as a basis turned out to be integrated into a more complex system, now it is covered by additional feedback loops. However, we found that this did not break the main principle of its functioning during the alternation of sleep and wakefulness phases. Thus, our model does not cancel the previously obtained results on this topic, but allows for their expanded interpretation.

2. The model structure we proposed can be easily applied to modeling a wide variety of scenarios for deviations in the sleep-wake rhythm. Above, in the Modeling of Nap Scenarios subsection, we gave one of the most obvious examples. However, specifying various combinations of neural activity parameters and PSD time dynamics opens up scope for targeted modeling of a wide variety of situations that potentially disrupt sleep. For example, this could be urgent night work, changes in the sleep schedule in the absence of darkness in polar regions, and much more.

If we talk about the limitations of our study, then first of all it is a very basic representation of the interaction pathways of the central sleep-wake switch and the periphery. However, this is also the focus of future modeling studies. In fact, this is one of the directions for developing the model as a whole - to clarify and include in the model the dynamic properties (time delays, nonlinearities) of wake drive and sleep drive signals along the path from the periphery to the central sleep-wake switch.

Another limitation is that we do not reproduce the physiological meaning of short restorative sleep, since the two-process model we used does not divide sleep into phases and since the nature of the restorative effect of short sleep is still under discussion, although it is not subject to doubt as a fact. In our model, the idea is implicitly conveyed that short sleep provides rest to the very functional network that was overloaded (local sleep). But this is a question that requires separate and detailed development.

As mentioned in the Introduction, we do not account for the existence of REM and NREM sleep states, focusing on deep (NREM) sleep only. Since REM sleep is characterized by high levels of neural activity, it contributes little (if anything) to accelerating the clearance of harmful metabolites and does not fit with our current working hypothesis. In our view, a different organizational level of model construction is required to adequately incorporate REM-NREM transitions. This could presumably be achieved by identifying typical spatial patterns of activity during REM sleep and associating them with various "medium-sized" sleep modules. Currently, the authors are unaware of experimental data that would allow for the valid construction of such a model.

Further validation of the model is desirable at various levels. Specifically, at the level of an individual NGVU, it is necessary to clarify the nature of state changes (e.g., the fine details of gliotransmitter dynamics) during overconsumption of metabolic reserves, such as astrocytic glycogen. At the system level, further validation of the model would highly require objective data on brain activity patterns in simulated situations that activate and overload as few neural structures as possible. This direction is the most promising, as it also offers potential for practical applications. Just as the two-process model was adapted for purely practical tasks on a daily time scale [63–65], our model may also be useful for studying the more rapid processes of drowsiness episodes during prolonged periods of concentration (long night drives, dispatchers working at a control panel, etc.).

Another important potential application is the use of the model as one of the tools in a comprehensive study of the possibility of non-simultaneous (local, sequential) activation of brain drainage mechanisms. It has already been shown that the phenomenon of local sleep is associated with local accelerated recovery after cognitive load [15]. It is conceivable that during prolonged sleep deprivation, such local events could serve to protect the brain and, to a certain extent, replace global sleep. Of course, experimental work is crucial here, but a suitable modeling paradigm is also important.

It has now been reliably shown that activation of the movement of cerebrospinal and interstitial fluids in the sleeping brain is associated with changes in the volume of astrocytes and the size of perivascular spaces [4,66]. There are pilot results indicating the involvement of $Ca^{2+}$-channels in the processes of changing the state of sleep and wakefulness, which is also accompanined by astrocytic regulation of neuronal activity of the brain during the transition from wakefulness to REM and NREM sleep [66].

During deep sleep, the size of perivascular spaces increases by 65%. This facilitates the removal of metabolites from the brain, such as beta-amyloid, which is produced by neurons daily and with high intensity [4]. It is known that metabolites are excreted from the brain precisely in deep sleep, when the drainage of its tissues is activated, in the regulation of which the meningeal lymphatic vessels play an important role[7,30]. However, due to the restriction of the technologies of visualizing the scenarios of the sleeping brain in real time, especially in humans, a detailed study of these processes is much difficult. Therefore, the appearance of models describing and clarifying the processes of brain drainage would contribute to navigation of research areas and the promotion of fundamental knowledge about the mechanisms of brain cleaning of metabolites and toxins. Since the deficiency or impaired sleep underlies a number of brain pathologies, including Alzheimer's disease. The use of our model in the research of the brain drainage could help in the development

 

of innovative technologies for controlling removal of toxins from the brain and predicting the therapeutic effectiveness of pathologies associated with suppression of brain drainage.

Regarding the quantitative development of the model, there are at least two approaches focused on different goals. Specifically, at the level of cellular structures, there are known studies on quantitative modeling of processes in NGVU [56,67–72]. This makes it possible to quantify molecular transport in the parenchyma.

However, the authors believe that accurate quantitative timing of the local compartments of the model is a more attractive and achievable goal, similar to what was done for the two-process model. The main challenge here is to accurately estimate the characteristic timing of processes such as various signaling pathways, for example, from parenchymal astrocytes to centers that sense sleep propensity.

## Supporting information

**S1 Appendix. The sleep-wake switch model.**
(PDF)

## Author contributions

**Conceptualization:** Dmitry Postnov, Oxana Semyachkina-Glushkovskaya, Jürgen Kurths.

**Funding acquisition:** Jürgen Kurths.

**Methodology:** Ksenia Merkulova.

**Project administration:** Jürgen Kurths.

**Software:** Dmitry Postnov, Ksenia Merkulova.

**Supervision:** Dmitry Postnov, Oxana Semyachkina-Glushkovskaya.

**Writing – original draft:** Dmitry Postnov, Ksenia Merkulova.

**Writing – review & editing:** Oxana Semyachkina-Glushkovskaya, Jürgen Kurths.

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
