## [Decision Letter · Decision Letter 0]

3 Nov 2025

PONE-D-25-46278Activity Driven Sleep Dynamics: a Conceptual Modeling StudyPLOS ONE

Dear Dr. Postnov,

Thank you for submitting your manuscript to PLOS ONE. After careful consideration, we feel that it has merit but does not fully meet PLOS ONE’s publication criteria as it currently stands. Therefore, we invite you to submit a revised version of the manuscript that addresses the points raised during the review process.

We look forward to receiving your revised manuscript.

Kind regards,

Gennady S. Cymbalyuk, Ph.D.

Academic Editor

PLOS ONE

Journal Requirements:

“DP and KM were supported by  Russian Science Foundation,  grant   25-15-00174.

OSG and JK were supported by  Russian Science Foundation,   grant 23-75-30001.”

Additional Editor Comments:

The authors should expand the discussion to outline possible applications, validation strategies, and quantitative development pathways. Limitations should be more explicitly addressed, particularly regarding the model’s lack of sleep stage differentiation and its implications for brain clearance mechanisms. In addition, minor language corrections and clearer labeling of figures (particularly the x-axes in Figures 2, 4, 6, and 7) are required to improve readability and precision.

Reviewer's Responses to Questions

**Comments to the Author**

1. Is the manuscript technically sound, and do the data support the conclusions?

Reviewer #1: Yes

Reviewer #2: Yes

2. Has the statistical analysis been performed appropriately and rigorously?

Reviewer #1: N/A

Reviewer #2: N/A

3. Have the authors made all data underlying the findings in their manuscript fully available?

Reviewer #1: Yes

Reviewer #2: No

4. Is the manuscript presented in an intelligible fashion and written in standard English?

Reviewer #1: Yes

Reviewer #2: Yes

5. Review Comments to the Author

Reviewer #1: The authors study a two population bistable model that switches between sleep and wake states, and includes SCN input and homeostatic drive, derived in prior work. In this work the authors incoroporate a psycho-sensory drive to simulate the effects of internal states and external influences on the system. They also incorporate distributed individual sleep units that can enter sleep-like states and that interact with the central sleep-wakefulness regulator; they use this framework of interaction between distributed sleep units and the central regulator to model naps. While there is a lack of detailed biological data on the functioning of individual sleep units distributed throughout the brain, and their interaction with the central sleep-wake switch, the authors provide an interesting and reasonable hypothesis through modeling of the nature of this dynamical system. A language edit may be useful prior to publication. I have found the following language errors/typos:

Line 6: “extracellular fluid”

Line 55: “Kruger says...” What does he say? Rewrite sentence

Line 61: On what time scale?

Line 78: Remove “be”

Line 303: how efficient the proposed structures (are)

Line 483 “substantial” should be “substantially”

Additionally, for Figure 2,4,6,7 the x axis should be clearly labeled (what are t and h?)

Reviewer #2: The authors present a novel mathematical model that integrates the central sleep-wake switch with the concepts of a "psycho-sensory drive" (PSD) and distributed neuro-glia-vascular "sleep units". They showed that the model reproduced the phenomena of local sleep and short nap episodes, which lie beyond the scope of traditional global sleep models. This approach is timely and addresses a significant gap in the computational neuroscience of sleep, offering a promising theoretical foundation for future, more detailed investigations.

The manuscript is well-structured and will be of interest to researchers in sleep science, computational neuroscience, and mathematical biology. To further strengthen the work, the following points should be addressed:

Major

1. The model is currently presented as a conceptual proof-of-principle. To enhance its impact, the authors should show its potential applications and the pathways for its future validation and quantitative development.

2. The discussion of limitations is good but could be expanded to include, for example, the model's current lack of sleep stage differentiation (NREM/REM) and more concrete hypotheses for its application to brain clearance mechanisms.

Minor

1. The manuscript requires minor language edits. For example, Line 79, “in the spirit of” what?

Line 210, missing “is” after “D”; Line 257, “Level” =>” level”; Line 532, “Ca2+” =>” Ca^2+” .

6. PLOS authors have the option to publish the peer review history of their article (what does this mean?). If published, this will include your full peer review and any attached files.

Reviewer #1: No

Reviewer #2: No

---

## [Author Response · Author response to Decision Letter 1]

15 Dec 2025

We are grateful to the reviewers for their positive assessment of our study and helpful suggestions for improvement.  Below are our responses to each of the reviewers' comments.

All changes made to the manuscript are highlighted in blue.

Reviewer #1:

Line 6: “extracellular fluid”

Our response: It was rephrased: “Thus, it has been shown that during deep sleep, the volume of  extracellular space  increases significantly.”

Line 55: “Kruger says...” What does he say? Rewrite sentence

Our response: It was rephrased as follows: “In the paper \cite{krueger2023tripping} the author expresses the opinion that "sleep is a property and is initiated by small groups of closely interconnected neurons."

Line 61: On what time scale?

Our response: It was rephrased: “From a temporal perspective, the main challenge for a two-process model is that it does not assume ''extra'' short episodes of sleep. “

Line 78: Remove “be”

Our response: It has been done

Line 303: how efficient the proposed structures (are)

Our response:  It was rephrased: “Since we propose the model structure from general considerations, although taking into account physiology, a numerical experiment is necessary to evaluate the effectiveness of the proposed structures.”

Line 483 “substantial” should be “substantially”

Our response: It was corrected.

Additionally, for Figure 2,4,6,7 the x axis should be clearly labeled (what are t and h?)

Our response: The x-axis in Figures 2,4,6,7 has been clarified (also in one of the graphs in Figure 5).

The designation "t, h" has been changed to "Time, hours".

Reviewer #2:

Major

1. The model is currently presented as a conceptual proof-of-principle. To enhance its impact, the authors should show its potential applications and the pathways for its future validation and quantitative development.

Our response:

Further validation of the model is desirable at various levels. Specifically, at the level of an individual NGVU, it is necessary to clarify the nature of state changes (e.g., the dynamics of gliotransmitter release) during overconsumption of metabolic reserves, such as astrocytic glycogen.

At the system level, further validation of the model would highly require objective data on brain activity patterns in simulated situations that activate and overload as few neural structures as possible. This direction is the most promising, as it also offers potential for practical applications. Just as the two-process model was adapted for purely practical tasks on a daily time scale,

our model may also be useful for studying the more rapid processes of drowsiness episodes during prolonged periods of concentration (long night drives, dispatchers working at a control panel, etc.).

Another important potential application is the use of the model as one of the tools in a comprehensive study of the possibility of non-simultaneous (local, sequential) activation of brain cleaning. It has already been shown that the phenomenon of local sleep is associated with local accelerated recovery after cognitive load.

It is conceivable that during prolonged sleep deprivation, such local events could serve to protect the brain and, to a certain extent, replace global sleep. Of course, experimental work is crucial here, but a suitable modeling paradigm is also important.

Regarding the quantitative development of the model, there are at least two approaches focused on different goals. Specifically, at the level of cellular structures, there are known studies on quantitative modeling of processes in NGVU. This makes it possible to quantify molecular transport in the parenchyma.

However, the authors believe that accurate quantitative timing of the local portion of the model is a more attractive and  achievable goal, similar to what was done for the two-process model. The main challenge here is to accurately estimate the characteristic timing of processes such as various signaling pathways, for example, from parenchymal astrocytes to centers that sense sleep propensity.

We have included everything described above in the Discussion text.

2. The discussion of limitations is good but could be expanded to include, for example, the model's current lack of sleep stage differentiation (NREM/REM)   and  more concrete hypotheses for its application to brain clearance mechanisms.

Our response:

We are grateful to the reviewer for this very pertinent comment. We indeed failed to address this issue in the previous version of the manuscript.

In the revised manuscript, we have supplemented the Introduction as follows:

Note that there is a general consensus that activation of harmful metabolite clearance during sleep is due to an increase in the proportion of extracellular space, which facilitates the transport of any molecules within the parenchyma. Since this increase occurs at low neural activity, all our considerations assume a state of deep sleep, NREM sleep, when SWA is recorded in the electroencephalogram.

We have also expanded the Discussion as follows:

As mentioned in the Introduction, one limitation of the model is that it does not distinguish between REM and NREM sleep states. Note that this is not possible in the original model of the two processes, which we supplement with "low-level" components.

Specifically, since REM sleep is characterized by high levels of neural activity, it contributes little (if anything) to accelerating the clearance of harmful metabolites and does not fit with our current working hypothesis. In our view, a different organizational level of model construction is required to adequately incorporate REM-NREM transitions. This could presumably be achieved by identifying typical spatial patterns of activity during REM sleep and associating them with various "medium-sized" sleep modules. Currently, the authors are unaware of experimental data that would allow for the valid construction of such a model.

Minor

1. The manuscript requires minor language edits. For example,

Line 79, “in the spirit of” what?

Our response: It was rephrased: “Next, we introduce a psycho-sensory drive signal into the model following the ideas expressed in the work ~\cite{george2018psycho}. “

Line 210, missing “is” after “D”;

Line 257, “Level” =>” level”;

Line 532, “Ca2+” =>” Ca^2+”.

Our response: It has been done.

---

## [Decision Letter · Decision Letter 1]

5 Jan 2026

Activity Driven Sleep Dynamics: a Conceptual Modeling Study

PONE-D-25-46278R1

Dear Dr. Postnov,

We’re pleased to inform you that your manuscript has been judged scientifically suitable for publication and will be formally accepted for publication once it meets all outstanding technical requirements.

Kind regards,

Gennady S. Cymbalyuk, Ph.D.

Academic Editor

PLOS One

Additional Editor Comments (optional):

Reviewers' comments:

Reviewer's Responses to Questions

**Comments to the Author**

1. If the authors have adequately addressed your comments raised in a previous round of review and you feel that this manuscript is now acceptable for publication, you may indicate that here to bypass the “Comments to the Author” section, enter your conflict of interest statement in the “Confidential to Editor” section, and submit your "Accept" recommendation.

Reviewer #1: All comments have been addressed

Reviewer #2: All comments have been addressed

2. Is the manuscript technically sound, and do the data support the conclusions?

Reviewer #1: Yes

Reviewer #2: Partly

3. Has the statistical analysis been performed appropriately and rigorously?

Reviewer #1: N/A

Reviewer #2: N/A

4. Have the authors made all data underlying the findings in their manuscript fully available?

Reviewer #1: Yes

Reviewer #2: No

5. Is the manuscript presented in an intelligible fashion and written in standard English?

Reviewer #1: Yes

Reviewer #2: Yes

6. Review Comments to the Author

Reviewer #1: (No Response)

Reviewer #2: The authors have addressed all my comments. I have no further questions. I believe the manuscript is now substantially strengthened.

7. PLOS authors have the option to publish the peer review history of their article (what does this mean?). If published, this will include your full peer review and any attached files.

Reviewer #1: No

Reviewer #2: No

---

## [Editor Report · Acceptance letter]

PONE-D-25-46278R1

PLOS One

Dear Dr. Postnov,

I'm pleased to inform you that your manuscript has been deemed suitable for publication in PLOS One. Congratulations! Your manuscript is now being handed over to our production team.

Kind regards,

on behalf of

Dr. Gennady S. Cymbalyuk

Academic Editor

PLOS One